# A Jasmonate-Responsive ERF Transcription Factor Regulates Steroidal Glycoalkaloid Biosynthesis Genes in Eggplant

**DOI:** 10.3390/plants11233336

**Published:** 2022-12-01

**Authors:** Tsubasa Shoji, Kazuki Saito

**Affiliations:** RIKEN Center for Sustainable Resource Science, Tsurumi-ku, Yokohama, Kanagawa 230-0045, Japan

**Keywords:** eggplant, ethylene, jasmonate, steroidal glycoalkaloids, transcription factor

## Abstract

Steroidal glycoalkaloids (SGAs) are a class of cholesterol-derived anti-nutritional defense compound that are produced in species of the genus *Solanum*, such as tomato (*S. lycopersicum*), potato (*S. tuberosum*), and eggplant (*S. melongena*). However, the regulation of defense-related metabolites in eggplant remains underexplored. In tomato and potato, the JASMONATE-RESPONSIVE ETHYLENE RESPONSE FACTOR 4 (JRE4) transcription factor positively regulates a large number of genes involved in SGA biosynthesis. Here, we report that the overexpression of eggplant *JRE4* (*SmJRE4*) induces numerous metabolic genes involved in SGA biosynthesis in leaves. We demonstrate the jasmonate-dependent induction of *SmJRE4* and its downstream metabolic genes and show that ethylene treatment attenuates this induction. Our findings thus provide molecular insights into SGA biosynthesis and its regulation in this major crop.

## 1. Introduction

The Solanaceae family contains around 2700 plant species that have adapted to a wide range of environments and are grown for food and other uses. Eggplant (*Solanum melongena*) is a widely cultivated vegetable crop of the Solanaceae family bearing edible fleshy fruits. While most plants in the family, including major crops such as tomato (*Solanum lycopersicum*) and potato (*Solanum tuberosum*), are indigenous to the Americas, eggplant is believed to originate from the Old World and was likely domesticated in and around the Indian subcontinent and China [1]. Eggplant is a representative species of the subgenus *Leptostemonum* (spiny *Solanum*), the largest monophyletic group in the plant family, with over 350 species [2].

Most species of the genus *Solanum* produce a class of cholesterol-derived specialized metabolites known as steroidal glycoalkaloids (SGAs), which accumulate as defense compounds that are toxic to a wide range of herbivores [3,4]. The two main SGAs found in eggplant, **α**-solasonine and **α**-solamargine, are generated through the glycosylation of a spirosolane-type aglycone solasodine featuring an F-ring nitrogen atom (N) in the **β** orientation at C-22. This differs from similar, but distinct, aglycones for tomato SGAs, which have the N in an opposite, **α** orientation in the F-ring [4] (Appendix A). This suggests a similarity in biosynthetic pathways for structurally related SGAs in tomato vs. eggplant, except for steps involved in F-ring formation (Appendix A).

Many SGA biosynthesis genes have been identified, mainly in tomato and potato [3]. In tomato and potato, the ETHYLENE RESPONSE FACTOR (ERF) family transcription factor JASMONATE-RESPONSIVE ERF 4 (JRE4), also known as GLYCOALKALOID METABOLISM 9 (GAME9), coordinates the transcription of numerous SGA biosynthesis genes, including those in the upstream primary mevalonate pathway [5,6,7]. In tomato, the defense-related plant hormone jasmonate (JA) induces the expression of *JRE4* and its downstream SGA pathway genes [6], and this JA-dependent induction is suppressed by ethylene treatment [7].

The molecular studies of SGA biosynthesis and its regulation in eggplant have largely lagged behind those in tomato and potato. Here, we characterize a number of SGA biosynthesis genes identified in the eggplant genome, including the transcription factor gene *SmJRE4*, which is related to tomato JRE4 (SlJRE4). We show that *SmJRE4* overexpression upregulates many genes for SGA metabolism and demonstrate JA-mediated gene induction and its attenuation by ethylene. We discuss the molecular basis of SGA biosynthesis in eggplant in comparison to that elucidated in tomato and potato.

## 2. Results and Discussion

### 2.1. Identification of a Group of Eggplant ERF Transcription Factors Related to SlJRE4

A group of six ERF family proteins predicted in the eggplant genome [8] are structurally related to SlJRE4 from tomato. To examine the phylogenetic relationships of the six ERFs from eggplant, SlJRE1 to SlJRE6 from tomato [6], and NtERF189 from tobacco (*Nicotiana tabacum*), which regulates the biosynthesis of the defense-related compound nicotine in the Solanaceae species [9,10], we constructed a phylogenetic tree and included AtERF13 from Arabidopsis (*Arabidopsis thaliana*) as the outgroup (Figure 1A). At least five pairs of apparent homologous ERF proteins were recognized among tomato and eggplant sequences (Figure 1A). Based on homology to SlJRE4 and the response to JA (see below), we named SMEL001g0132170 eggplant JRE4 (SmJRE4). SMEL001g132160, SMEL001g133170, SMEL001g133180, SMEL001g133190, and SMEL005g225020 were named SmJRE4-like 1 (SmJRE4L1), SmJRE4L2, SmJRE4L3, SmJRE4L4, and SmJRE4L5, respectively (Figure 1). In the eggplant genome, SmJRE4L5 is present as a single copy that is not in a cluster with other *JRE* genes, whereas *SmJRE4* and other *SmJRE4L* genes form two distinct clusters that are separated by 1584 kb on the chromosome I. One cluster includes *SmJRE4* and *SmJRE4L1*, and another cluster includes *SmLRE4L2*, *SmJRE4L3*, and *SmJRE4L4* (Figure 1B). The pattern of gene organization suggests that a chromosomal rearrangement disrupting a possible ancestral five-gene cluster occurred in the lineage leading to eggplant.

### 2.2. Putative SGA and Related Phytosterol Biosynthesis Genes in Eggplant

To generate an inventory of putative genes involved in SGA and related phytosterol biosynthesis pathways in eggplant (presuming the overlap of tomato and eggplant pathways; see above), a BLASTP search was performed using a set of tomato proteins (Appendix A) as queries against the proteins predicted in the eggplant genome (eggplant genome consortium ver. 3) [8] with cut-off values of amino acid identities > 65% and e-values < 1 × 10^−100^. We then excluded the genes that were expressed at very low levels (maximum RPKM values < 1.0) [8]. In all, 44 metabolic genes were identified as putative SGA and phytosterol biosynthesis genes in eggplant (Figure 2, Appendix A). As reported already [3,8], *SmGAME2* (SMEL007g282430) and *SmGAME11* (SMEL007g282440) form a metabolic gene cluster in the eggplant genome, similar to their homologs in the tomato and potato genomes [3,11]. No *SmGAME* genes other than the two genes that had been reported to form gene clusters [3,8] were retrieved with our criteria.

Based on phylogenetic relationships to their tomato and potato counterparts (Appendix A) [12], we identified and defined eggplant genes for cholesterol biosynthesis—*SmSSR2* (SMEL000g071270), *SmSMO3* (SMEL001g131240), *SmSMO4* (SMEL006g247500), and *SmDWF7-2* (SMEL000g029670)—that have evolved through gene duplication and divergence from genes for phytosterol biosynthesis [12] (Figure 2, Appendix A). While tomato and potato have *DWF5-1* and *DWF5-2* specialized to phytosterol and cholesterol pathways, respectively [12], only *SmDWF5* (SMEL001g148750) is present in the eggplant genome (Appendix A); it could not be assigned to either *DWF5-1* or *DWF5-2* because its structural and enzymatic properties have remained elusive and may be shared between the two pathways (Figure 2).

Illumina-based expression data from the SGA and phytosterol pathway genes obtained previously [8] were visualized in a heat map (Appendix A). Hierarchical clustering did not generate clusters with seemingly unique expression patterns or any apparent clusters enriched with genes specialized in cholesterol and SGA biosynthesis. This result is in clear contrast to a similar analysis performed with a set of the tomato genes [7], possibly reflecting the fact that eggplant produces a certain amount of SGA in all organs, including mature fruits. Indeed, drastic decreases in RPKM values of stage 3 fruit relative to stage 1 fruit were demonstrated for only eight (>10-fold decrease) and two (>20-fold decrease) out of 44 genes examined. We also saw increased expression during maturation for some genes, such as *SmGAME4* (SMEL12g381730), implying that no overall marked decrease in gene expression associated with SGA biosynthesis happens during fruit maturation in eggplant. To better understand SGA metabolism during fruit maturation in eggplant, it is critical to comprehensively characterize the metabolites and expression of metabolic genes, including those mediating the conversion from bitter SGAs to non-bitter forms [13,14].

*SmJRE4* expression was high among the *ERF*s in the flowers and fruits at stage 1, though *SmJRE4* was not expressed in the fruits after stage 2 (Appendix A). Although expression was not detected for all five *ERF*s in the leaves, their transcripts were observed in the roots, where the expression of *SmJRE4* was not necessarily predominant, compared to the *SmJRE4L*s (Appendix A). Such expression of *SmJRE4* in the roots appears to be exceptional, considering the predominant expression of *SlJRE4* [6] and *NtERF189* [15] in nearly all organs in their respective species.

### 2.3. Induced Expression of SGA Biosynthesis Genes by Overexpression of SmJRE4

To examine whether SmJRE4 regulates SGA biosynthesis genes in eggplant, we used the *Agrobacterium*-mediated transient overexpression of *SmJRE4* and related *ERF*s, *SlJRE4* from tomato and *NtERF189* from tobacco, performed under the control of the cauliflower mosaic virus (CaMV) 35S constitutive promoter in the leaves of 4-week-old eggplant. Transcript levels of select SGA and phytosterol pathway genes (indicated in Figure 2 and Appendix A) were determined 2 d after agroinfiltration by reverse transcription-quantitative PCR (RT-qPCR) analysis. We analyzed two genes involved in steps before cycloartenol (*SmHMGR2* [SMEL002g153730] and *SmCAS2* [SMEL004g212830]), many of the genes contributing to SGA production after cycloartenol (purple and red genes indicated in Figure 2), and three phytosterol pathway–specific genes (*SmSMT1* [SMEL001g137920], *SmDWF7-1* [SMEL002g159860], and *SmSSR1* [SMEL002g157630]). When there were multiple genes for a particular enzymatic step, the genes with higher expression levels were chosen. Note that the possibility of nonspecific amplification of related sequences could not be excluded, as the specificities of primers were not validated experimentally.

The overexpression of the transcription factor genes was confirmed by RT-qPCR analysis (Appendix A). When either *SmJRE4*, *SlJRE4*, or *NtERF189* was transiently overexpressed, significant increases in expression relative to the controls were observed. These increases typically occurred at similar levels among the introduced *ERF*s (in a range of 4.0- to 48.8-fold for *SmCAS2* and all its downstream genes contributing to SGA production), including those shared with the phytosterol pathway, except for *SmHSD2* (SMEL002g157360) and *SmDWF5* (Figure 3). The increases from *NtERF189* overexpression were significantly smaller for *SmSMO3*, *SmSMO4*, and *SmDWF7-2* than those from *SmJRE4* or *SlJRE4* overexpression (Figure 3). In contrast, no induction of phytosterol-specific *SmSMT1*, *SmDWF7-1*, and *SmSSR1*, as well as the mevalonate pathway gene *SmHMGR2*, occurred as a result of *ERF* overexpression. This indicates that the ERF-mediated transcriptional activation observed is specific to the pathway leading to SGAs (Figure 3). It remains largely undetermined whether genes encoding enzymes of early steps (including those in the mevalonate pathway, which usually form families of multiple members [Figure 2, Appendix A]) are regulated by SmJRE4.

In contrast to clear gene induction by ERFs from different species in eggplant leaves (Figure 3), SlJRE4, but not NtERF189, induced the promoters of SGA biosynthesis genes in tomato fruits [7], suggesting that more specific factors are required to regulate SGA production in tomato than in eggplant. It is also worth noting that, as demonstrated here (Figure 3, Appendix A), eggplant leaves are easily amenable to transient overexpression assay in contrast to tomato and potato leaves.

### 2.4. Responses of SmJRE4 and Its Downstream SGA Biosynthesis Genes to JA and Ethylene

To understand the defensive properties of toxic specialized metabolites, especially in comparison to those in related species [6,7,16,17], we examined whether and how SGA biosynthesis genes are regulated by JA and ethylene in eggplant. For the treatments, 4-week-old eggplant plants were exposed to gases of methyl jasmonate (MJ) and ethylene in air-tight plastic containers (Appendix A). The ripening agent Uregoro (Carto, Shizuoka, Japan), which consists of ethylene-adsorbed zeolite, was used for the stable and easy release of ethylene gas. The transcript levels in the leaves from the plants subjected to the various treatments were determined by RT-qPCR.

We first measured the expression of *SmJRE4* and the SGA biosynthesis genes *SmSSR2* and *SmGAME2* from 0 to 48 h following treatment with MJ. *SmJRE4*, *SmSSR2*, and *SmGAME2* were gradually induced by MJ, with peaks at 24 h (5.2- to 9.9-fold relative to the levels at 0 h). Significant increases were first detected at 15 h for all three genes; the induction of *SmJRE4* was not earlier than those of the downstream genes (Figure 4). The gradual induction of a transcriptional regulator gene nearly in parallel with its downstream genes has also been reported for *SlJRE4* [6] and *NtERF189* [9], whereas JA more acutely and strongly induces most *ERF*s that are homologous to them but presumably are not involved in regulating the defense metabolism in each species [6,9].

To investigate the gene responses, 4-week-old eggplant plantlets were treated with MJ, ethylene, or both for 24 h, and transcript levels of *SmJRE4* and a series of its target and off-target metabolic genes were analyzed by RT-qPCR in leaves (Figure 5). *SmJRE4* and most of its downstream genes, but no off-target ones, were significantly induced by MJ (2.4- to 8.7-fold relative to mock-treated controls, Figure 5). The JA-dependent induction of these genes was clearly diminished when ethylene was applied simultaneously (Figure 5). Apart from the induction of *SmHMGR2*, ethylene alone did not cause the induction of any genes, but instead caused a significant 1.4- to 6.8-fold decrease in the expression of nine metabolic genes (Figure 5). Overall, we demonstrated both the JA-elicited expression of *SmJRE4* and numerous SmJRE4-regulated metabolic genes and its suppression by ethylene (Figure 4 and Figure 5). These results suggest that responses to the phytohormones are a conserved property that is likely fundamental among SmJRE4 and related ERF transcription factors and their target defense metabolic pathways in plants of the Solanaceae family [7,17]. It will be intriguing to address the biological significance of the antagonistic interplay between defense-related JA and ethylene signals controlling defenses in ecological contexts [18].

In conclusion, we found that the JA-responsive ERF transcription factor SmJRE4 up-regulates numerous SGA biosynthesis genes and that ethylene suppresses the JA-dependent induction of most of the SmJRE4-regulated genes in the leaves of eggplant. These results provide insight into the regulatory function of SmJRE4 in defense-related SGA metabolism in eggplant, a major crop in the Solanaceae family. Genetic manipulation of SGA production exploiting a transcriptional regulator SmJRE4 contributes to crop improvement in terms of plant protection against predators. This study is another example of ERF transcription factors involved in regulating defense metabolites in plants and reveals the functional conservation and divergence of structurally related ERFs [10].

## 3. Materials and Methods

### 3.1. Plant Growth and Treatment

Seeds of eggplant, *Solanum melongena* cv. PC Chikuyoh, were obtained from Takii & Co., Ltd. (Kyoto, Japan). Sterilized seeds were germinated on half-strength Gamborg B5 medium solidified with 0.7% (*w*/*v*) agar and supplemented with 3% (*w*/*v*) sucrose. Two-week-old seedlings were transferred to soil (JA Nipi Horticulture Soil No. 1; Nippon Hiryo, Gunma, Japan) in pots and grown in a culture room at 25 °C in 16 h-light/8 h-dark conditions.

Four-week-old plants were exposed to methyl jasmonate (MJ) and/or ethylene gases in 12-L air-tight plastic buckets (Appendix A). A piece of paper towel absorbed with 1.2 mL of a volatile liquid form of MJ (Fujifilm, Tokyo, Japan) and/or two needle-pierced packs of the ripening agent Uregoro (Carto, Shizuoka, Japan), which contains ethylene-adsorbed zeolite, were placed along with two individual plants in a bucket for MJ and ethylene treatments, respectively. According to the manufacturer, approximately 11 mL ethylene gas is released from one pack of Uregoro for 60 min after piercing.

### 3.2. Phylogenetic Analysis

Amino acid sequences of the full-length proteins were aligned using MUSCLE [19]. An unrooted phylogenetic tree was constructed using the neighbor-jointing algorithm with MEGAX [20] with default settings.

### 3.3. Agrobacterium-Mediated Transient Overexpression

The full-length coding region of *SmJRE4* (SMEL001g132170) was amplified by PCR using a cDNA template, which was prepared from 3-week-old whole plants. The PCR amplicon was cloned into pENTR/D-TOPO (Thermo Fisher Scientific, Waltham, MA, USA) and sequenced to confirm that no mutations were introduced during PCR. The sequence was inserted into a binary vector pGWB2 [21] through a reaction catalyzed by LR Clonase II (Thermo Fisher Scientific), which placed it under the control of the cauliflower mosaic virus (CaMV) 35S promoter. The pGWB2-based vectors for the overexpression of *SlJRE4* (Solyc01g090340) and *NtERF189* (GenBank: AB827951) [7,9], the control vector pBI121 (GenBank: AF485783), and the p19 vector for silencing suppression [22] were used. Vectors were introduced into *Agrobacterium tumefaciens* strain GV3101 with heat shock. The transcription factor genes were transiently overexpressed in the youngest fully expanded leaves of 4-week-old eggplant plants as described [23]. Bacterial suspensions of the vector for overexpression of each transcription factor gene or the control vector and the p19 vector [22] were combined in a ratio of 7:3 by volume. Gene expression was analyzed by RT-qPCR using leaves harvested 2 d following Agrobacterium infiltration.

### 3.4. Reverse Transcription-Quantitative PCR (RT-qPCR) Analysis

Plant samples were ground to a powder in liquid nitrogen, and the total RNA was isolated using a RNeasy kit (Qiagen, Venlo, The Netherlands). The RNAs were converted to first-strand cDNAs using SuperScript IV reverse transcriptase (Thermo Fisher Scientific) and an oligo (dT) primer. The cDNA templates were amplified using a StepOnePlus Real-Time PCR system (Thermo Fisher Scientific) and the Fast SYBR Green Master Mix (Thermo Fisher Scientific) with a thermal program described previously [24]. The cyclophilin gene SMEL001g116150 was used as a reference to normalize the data [25]. Primer sequences are listed in Appendix A.

## Figures and Tables

**Figure 1 plants-11-03336-f001:**
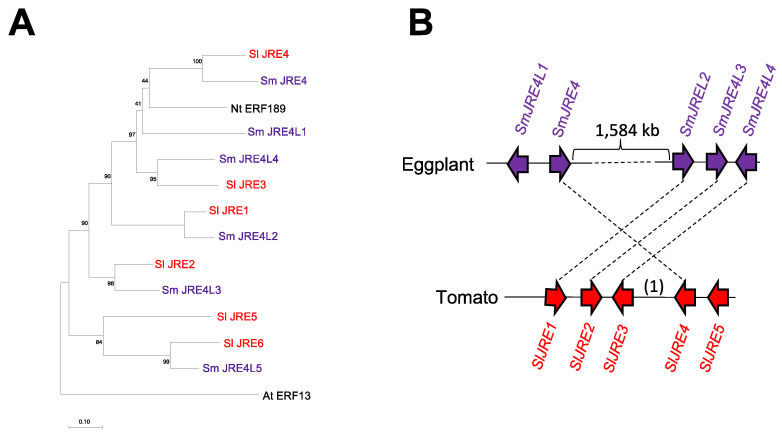
A group of ethylene response factor (ERF) transcription factors from eggplant related to tomato JASMONATE-RESPONSIVE ERF 4 (SlJRE4). (**A**) A phylogenetic tree created using sequences of eggplant (Sm; *Solanum melongena*) SmJRE4 and related proteins. SmJRE4 (SMEL001g132170), SmJRE4-like 1 (SmJRE4L1) (SMEL001g132160), SmJRE4L2 (SMEL001g133170), SmJRE4L3 (SMEL001g133180), SmJRE4L4 (SMEL001g133190), and SmJRE4L5 (SMEL005g225020); tomato (Sl; *Solanum lycopersicum*) SlJRE1 (Solyc01g090300), SlJRE2 (Solyc01g090310), SlJRE3 (Solyc01g090320), SlJRE4 (Solyc01g090340), SlJRE5 (Solyc01g090370), and SlJRE6 (Solyc05g050790); tobacco (Nt; *Nicotiana tabacum*) NtERF189 (GenBank: AB827951.1); and Arabidopsis (At; *Arabidopsis thaliana*) AtERF13 (At2g44840) were included in the analysis. The percentage support from 1050 bootstraps is indicated at the branch nodes. The scale bar indicates the number of amino acid substitutions per site. (**B**) Schematic diagrams of *ERF* gene clusters in the eggplant and tomato genomes. The orders and orientations of *ERF*s (arrowheads) are shown. Homologous genes are connected by broken lines. A gene not encoding an ERF (1) is found between *SlJRE3* and *SlJRE4* in the tomato *ERF* cluster.

**Figure 2 plants-11-03336-f002:**
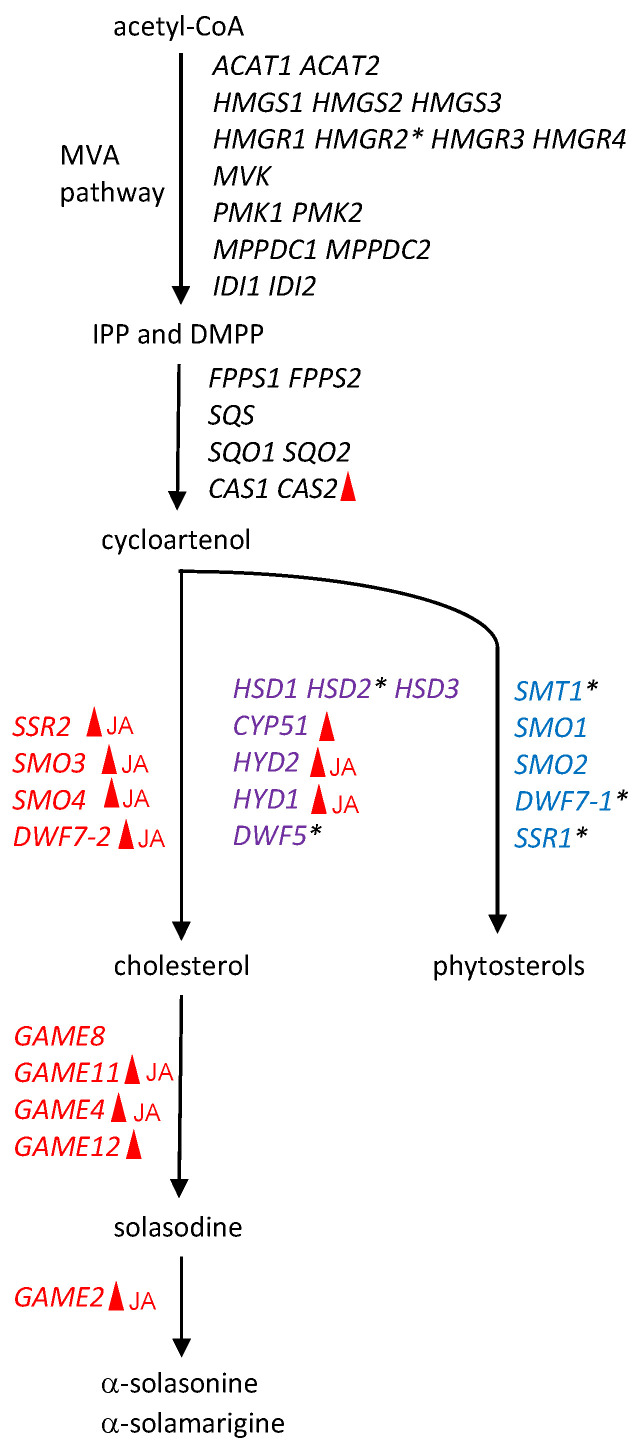
Schematic illustration of the SGA and related phytosterol biosynthesis pathways in eggplant. Based on homologies (amino acid identity > 65%, e-value < 1 × 10^−100^) to tomato counterparts (Appendix A) and the expression levels of their respective genes (maximum RPKM value > 1.0) [8], the proteins predicted in the eggplant genome (eggplant genome consortium v3) were deduced to be involved in SGA and related phytosterol pathways. The genes are listed in Appendix A and are presented along the pathways here. For steps after the common intermediate cycloartenol, the metabolic genes specific to SGA biosynthesis (red), phytosterol biosynthesis (blue), and those shared by both (purple) are shown. Details of the part of the pathway from cholesterol to solasodine are shown in Appendix A. Genes demonstrated to be regulated by any ERF (see Figure 3) are indicated with red arrowheads, and those induced by methyl jasmonate (see Figure 4 and Figure 5) are labeled JA. Genes examined in this study, but not regulated by either ERFs or methyl jasmonate, are marked with asterisks. MVA; mevalonate, IPP; isopentenyl pyrophosphate, DMPP; dimethyl pyrophosphate. Other abbreviations and identification numbers of the genes are presented in Appendix A.

**Figure 3 plants-11-03336-f003:**
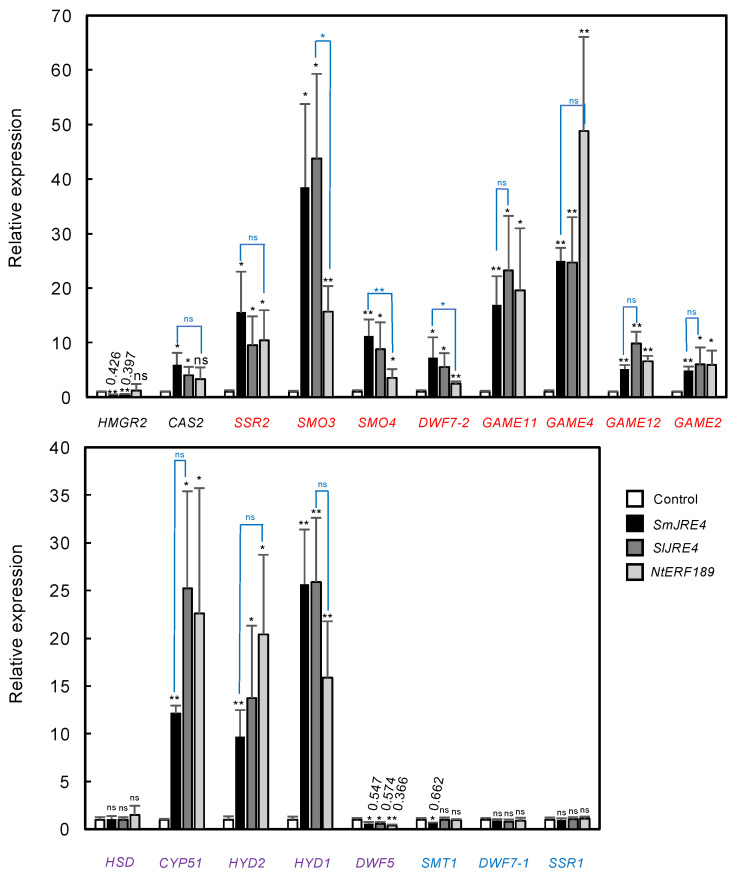
*Agrobacterium*-mediated transient overexpression of *SmJRE4* and the related genes *SlJRE4* and *NtERF189* induced the expression of multiple SGA biosynthesis genes in eggplant leaves. Fully expanded leaves in 4-week-old eggplant were infiltrated with Agrobacterium harboring a pGWB2-based binary vector that is designed to overexpress either *SmJRE4*, *SlJRE4*, or *NtERF189* under the control of the CaMV35S promoter. Transcript levels in the leaves 2 d after infection were determined by RT-qPCR analysis. The metabolic genes specific to SGA biosynthesis (red), phytosterol biosynthesis (blue), and those shared by both (purple) were analyzed. Primer sequences are listed in Appendix A. Expression data for the transcription factor genes are shown in Appendix A. Means and standard deviations of biological replicates (*n* = 3 or 4) are indicated. Expression levels are presented relative to those in the pBI121-transformed controls. Significant differences relative to the controls (black) and between the values indicated (blue) were determined by Student’s *t*-test. * *p* < 0.05; ** *p* < 0.01. ns; not significant.

**Figure 4 plants-11-03336-f004:**
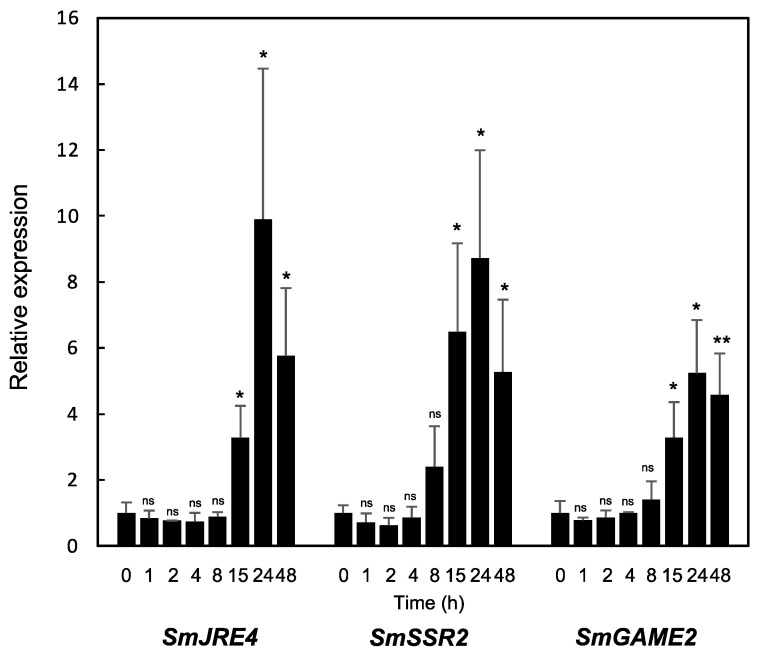
Jasmonate (JA)-induced expression of *SmJRE4* and SGA biosynthesis genes *SmSSR2* and *SmGAME2* in eggplant leaves. Four-week-old plants were exposed to methyl jasmonate for 0, 1, 2, 4, 8, 15, 24, or 48 h in an air-tight plastic bucket (see Appendix A). Transcript levels were determined by RT-qPCR analysis. Primer sequences are listed in Appendix A. Means and standard deviations (error bars) of three biological replicates are indicated. Significant differences of each treatment relative to the 0-h control samples were determined by Student’s *t*-test. * *p* < 0.05; ** *p* < 0.01. ns; not significant.

**Figure 5 plants-11-03336-f005:**
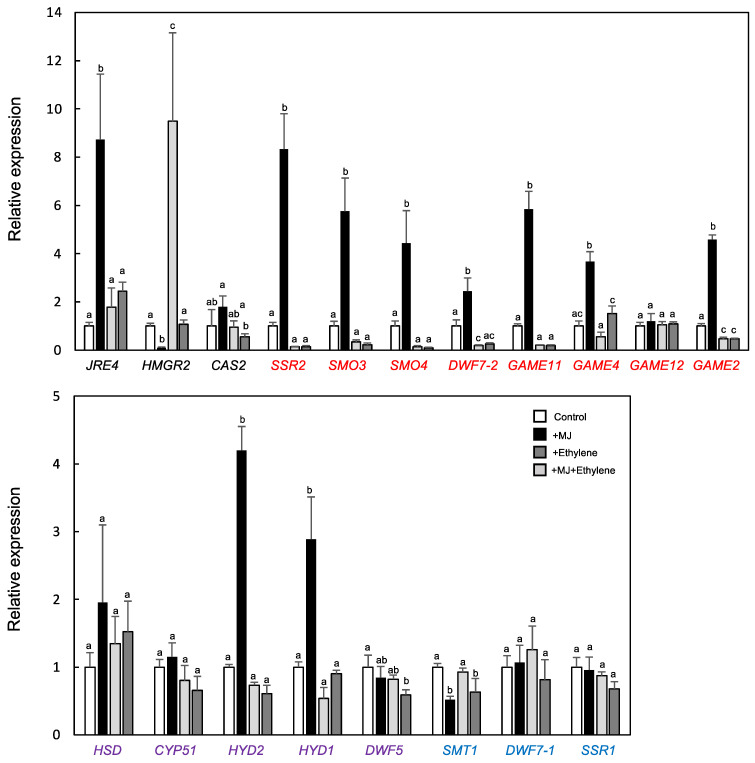
Relative expression of SGA biosynthesis genes in response to methyl jasmonate (MJ) and ethylene treatments in eggplant leaves. Four-week-old tomato plants were exposed to MJ (black), ethylene (light grey), or both (MJ + ethylene) (dark grey) for 24 h in an air-tight plastic bucket (see Appendix A). Transcript levels were determined by RT-qPCR analysis. Primer sequences are listed in Appendix A. The metabolic genes specific to SGA biosynthesis (red), phytosterol biosynthesis (blue), and those shared by both (purple) were analyzed. Means and standard deviations (error bars) of biological replicates (*n* = 3 or 4) are indicated. Different lowercase letters indicate significant differences between values at *p* < 0.05, as determined by one-way analysis of variance (ANOVA) followed by a Tukey–Kramer test.

## Data Availability

Not applicable.

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
