# Peer review of "A Jasmonate-Responsive ERF Transcription Factor Regulates Steroidal Glycoalkaloid Biosynthesis Genes in Eggplant"

_plants, 2022, doi:10.3390/plants11233336_

Round 1

Reviewer 1 Report

This is a solid work on role of jasmonate (JA)-responsive ERF transcription factors in regulation of steroidal glycoalkaloid (SGA) biosynthesis genes in eggplant. Even there is a remarkable amount of knowledge on ERFs and SGAs in tomato and potato, much less is known for the eggplant. The data are new and add additional knowledge on SGA biosynthesis for another Solanaceae species. More specifically, the paper is analyzing the positive regulation of SGA biosynthesis genes by the JASMONATE-RESPONSIVE ETHYLENE RESPONSE FACTOR 4 (JRE4), a key regulator, upon its overexpression. The initial phylogenetic analysis JRE 4 in relation to other JRE and other plants such Arabidopsis and tobacco revealed a chromosomal rearrangement (Fig. 1). Here, it was of advantage that both authors have long-term experience in analysis of ERFs, e.g. T. Shoji for nicotine biosynthesis.

The paper is a carefully performed analysis of JRE4 of eggplant plantlets (roots, leaves, flowers, fruit stages) in comparison to corresponding transcription factors of potato and tomato. Induction if SGA biosynthesis genes by jasmonates and its suppression by ethylene were clearly shown, and the consequences of overexpression of JRE4 give further insights on regulation of metabolic genes of SGA biosynthesis in eggplants. Expression data of SGA genes and phylogenetic relationships of proteins of the different Solanaceaen species were presented. Even the data are solid, the presentation can be improved.

The list of references has to be checked carefully: Give complete page number for ref. 2, 6, 8, 11, 12, 13, 14, 15, 16, 17, 18, 19, 22, 24.

Author Response

Reference part was carefully checked and amended.

As for presentation improvement, could you point the parts necessary to change? If so, we could respond point by point. Thank you much.

Reviewer 2 Report

Manuscript ID:  Plants-2048710

Title: A jasmonate-responsive ERF transcription factor regulates steroidal glycoalkaloid biosynthesis genes in eggplant

Dear Editor,

I went thoroughly on this current manuscript. After my observations this manuscript not suitable for the publication in the Plants journal. I strongly disagree to accept this manuscript. Reasons mentioned below.

1. The outcome from the experiments lacks novelty.

2. Authors mentioned that JRE4 TF positively regulates many genes involved in SGA biosynthesis. However, authors showed only qualitative data. There is no strong molecular data to support the outcomes.

3. The expression of results like methodology, results and discussion seriously needs to be improve.

4. English editing is necessary for this current version of manuscript; it contain many grammatical errors.

Author Response

The outcome from the experiments lacks novelty.

> As mentioned in Introduction, eggplant is phylogenically distant from tomato and potato within the genus Solanum and little had been known on transcriptional regulation of SGA biosynthesis genes. I believe the novelties  of our findings reach the standard of this journal.

Authors mentioned that JRE4 TF positively regulates many genes involved in SGA biosynthesis. However, authors showed only qualitative data. There is no strong molecular data to support the outcomes.

> In Fig. 3, up-regulation of at least 12 SGA biosynthesis genes by SmJRE4 were demonstrated with RT-qPCR analysis. RT-qPCR is considered quantitative in nature, I suppose. 

The expression of results like methodology, results and discussion seriously needs to be improve.

> Please mention more specifically. 

English editing is necessary for this current version of manuscript; it contain many grammatical errors.

> Please point some of the errors. The  manuscript was checked by a professional service PlantEditors (https://planteditors.com)

Author Response

Thank you for your suggestion.  SGAs are defense chemicals against a wide range of biotic predators. This study gives useful insights into molecular basis of SGA regulation in the important crop and thus advance our ability to manipulate the defense-related metabolic property. In the conclusion paragraph, a one sentence  was added to address the concern.

Round 2

Reviewer 2 Report

Dear Editor,

After my observation on revised version of this manuscript, I recommend this to accept in this current form.